# Anion effects govern efficiency of electrochemical amine-mediated $CO_2$ capture/release

Liang Liang [1,5], Frederik Firschke[1,5], Jie Wang[2,5], Li Yang [2] ✉, Xingli Wang [1], Wen Ju [1,3], Matthew T. Mayer [4] & Peter Strasser [1] ✉

Ambient electrochemical $CO_2$ capture powered by renewable energy offers a promising carbon removal route, exemplified by the emerging electrochemically mediated amine regeneration (EMAR) process demonstrated in lab-scale single cells and stacks. However, molecular-level insight into EMAR interfacial kinetics is still missing, particularly at the anode, where $CO_2$ release involves a mechanistically non-trivial re-complexation process at the electrode–electrolyte interface, coupling heterogeneous metal-ion release with bulk complexation. Here, we report the time-resolved characterization of the interfacial molecular processes of the EMAR $CO_2$ release process. Using in situ Fourier-transform infrared (FTIR) spectroscopy and ultraviolet-visible (UV-vis) spectroscopy, cyclic voltammetry, and real-time differential electrochemical mass spectrometry (DEMS), we examine how the nature of the electrolyte anion affects the $CO_2$ release onset potentials. The time-resolved analyses reveal that $Cl^-$ ions are more effective in releasing Cu ions and hence $CO_2$ than nitrate or perchlorate. Molecular dynamics simulations show that strong surface Cu−Cl interactions likely facilitate favorable $CO_2$ and carbamate adsorption kinetics. We expect that this study paves the way for broader use of interfacial in-situ analytics in electrified $CO_2$ capture and release.

Engineering carbon capture is poised to become a key technology to address global climate challenges, particularly targeting the capture and separation of $CO_2$ from industrial point sources and off-gases[1–4]. Unlike its conventional temperature and pressure swing analogues, electrochemical $CO_2$ capture, separation, and concentration strategies powered by clean electricity hold the promise to be a cheap, energy efficient, and scalable carbon capture technologies[5–7]. Among them, electrochemically mediated amine regeneration (EMAR) is emerging as an attractive electricity-based variant of more traditional thermal amine-based $CO_2$ capture/release technologies[8–10]. The EMAR desorption process is the electrochemical analogue (electrochemical swing) of the circular thermal approaches (thermal swing), with the

reduced-pressure high-temperature desorption stage replaced with a two-compartment electrochemical desorber cell that operates at the same low temperatures as the absorber[11].

The circular electrochemical reactions of the EMAR $CO_2$ capture and release cycle start with the adsorptive $CO_2$ capture by an amine (Am) solutions yielding a carbamate (Am-$CO_2$) solution

$$CO_2 + Am \rightarrow (Am - CO_2), K_{Am-CO2} \sim 10^3 - 10^5 \qquad (1)$$

The $K_{Am\text{-}CO2}$ value refers to the case of ethylenediamine (EDA). The carbamate solution exiting the absorber is fed to the anode compartment of a divided electrochemical EMAR cell for $CO_2$

[1]Chemical Engineering Division, Department of Chemistry, Technical University Berlin, Berlin, Germany. [2]Institutes of Physical Science and Information Technology, Anhui University, Anhui, China. [3]Department of Electrochemistry and Catalysis, Leibniz Institute for Catalysis, Rostock, Germany. [4]Electrochemical Conversion, Helmholtz-Zentrum Berlin für Materialien und Energie, Berlin, Germany. [5]These authors contributed equally: Liang Liang, Frederik Firschke, Jie Wang. ✉e-mail: yangli91@mail.ustc.edu.cn; pstrasser@tu-berlin.de

desorption and sorbent regeneration. At the anode, copper ions are electrochemically generated from a copper plate anode

$$Cu^0 \rightarrow Cu^{2+} + 2e^- \tag{2}$$

to drive the fast competitive complexation of Am-$CO_2$ carbamates into free $CO_2$ and a colored Cu-Am complex

$$Cu^{2+} + n(Am - CO_2) \rightarrow Cu(Am)_n^{2+} + nCO_2, K_{Cu(Am)n} \sim 10^{18} - 10^{20} \tag{3}$$

Combining the two latter reactions results in the effective anodic EMAR process in which metallic copper (Cu) undergoes a coupled redox/complexation reaction with Am-$CO_2$ to form $Cu(Am)_n^{2+}$ complexes[12–14]. The difference in the two equilibrium (i.e., stability) constants, $K_{Am\text{-}CO2}$ and $K_{Cu(Am)n}$, ensures fast and quantitative competitive release of $CO_2$.

The $CO_2$ is then separated from the liquid electrolyte, and the latter is fed to the cathode of the electrochemical EMAR cell where the amine sorbent is regenerated from the $Cu(Am)_n^{2+}$ complex by electrochemical Cu deposition

$$Cu(Am)_n^{2+} + 2e^- \rightarrow Cu + n\,Am \tag{4}$$

And the free amine fed back into the adsorber stage to close the EMAR cycle.

Generally, the electrochemical redox behavior of surface Cu species, such as metallic adatoms and Cu ions, during the interfacial $CO_2$ release (Eqs. 2 and 3) will be strongly influenced not only by the applied overpotential and the nature of the Cu surface, but also by the composition and possible adsorption of electrolyte molecules and ions at the electrochemical interface. In other words, given a electrochemical driving force, the nature and concentration of carbamate, free amine, dissolved gases, anions, cations, and solvent molecules will influence the onset potential and kinetics of the local interfacial generation of Cu ions from Cu[15–19]. For instance, electrolyte ions may form a blocking adlayer on the electrode surface, hindering the Cu ion generating redox reaction[20]. Alternatively, ions at the electrode surface may enhance the generation of Cu ions through strong ion-ion interactions, thereby shifting the redox reaction and Cu ion onset

potential[21,22]. Indeed, bulk electrolyte composition was reported to modulate the electric energy demand for $CO_2$ release and amine regeneration in the EMAR process[23]. However, most prior work on EMAR processes has essentially focused on electrolyte and device engineering aspects at the full single EMAR cell level. By contrast, time-resolved exploration and understanding of the elementary interfacial EMAR process steps at the molecular scale, in particular using advanced in-situ and operando analytical techniques, have not been conducted to date. Hence, the detailed molecular reaction kinetics along with the nature and origin of kinetic barriers of individual EMAR steps are poorly understood to date. We intend to fill this knowledge gap, as the electrocatalytic science community is moving into the research area of electrified $CO_2$ capture and concentration. A deeper understanding of the interfacial reaction kinetics will enable the design of structurally and energetically optimized electrochemical EMAR microenvironments that will bring to bear more efficient electrochemical $CO_2$ capture and concentration technologies[3,13,24].

Here, we report the discovery of a voltage-efficient $Cl^-$-mediated interfacial EMAR $CO_2$ release pathway that enables one of the lowest ever reported onset potentials of $CO_2$ release (Fig. 1)[23,25,26]. Using time-resolved, online and in situ spectrometry and spectroscopy, we unravel relations between the kinetic onset potentials of interfacial $CO_2$ release, closely linked to the generation of bulk $Cu^{2+}$ species, and the nature of the electrolyte anions. Combining our experiments with computational modeling, we discuss the trends in $CO_2$ release kinetics, with the interfacial adsorption of electrolyte species. Cu nanoparticles (NP) and smooth polycrystalline Cu surfaces in presence of supporting KCl electrolyte showed significantly less positive Cu redox onset potentials—as low as $-0.36\ V_{Ag/AgCl}$ – in 0.1 M $CO_2$-saturated- EDA-KCl (KCl + EDA-$CO_2$) bulk electrolytes compared to $KNO_3$ and $KClO_4$ electrolytes. Time-resolved Differential Electrochemical Mass Spectrometry (DEMS) confirmed this earlier onset to coincide with earlier $CO_2$ release. Complementary in situ FTIR and in situ UV-vis were used, respectively, to track the formation of $CO_2$ and the Cu-amine complex species at the interface. Moreover, the voltage and efficiency gains offered by the anion engineering in Cu-based systems were quantitatively evaluated, leading to significantly more energy savings. Molecular dynamics (MD) calculations provided a theoretical insights for our observations and suggested the anion-regulated molecular interactions at the Cu electrode. Our complementary experimental and

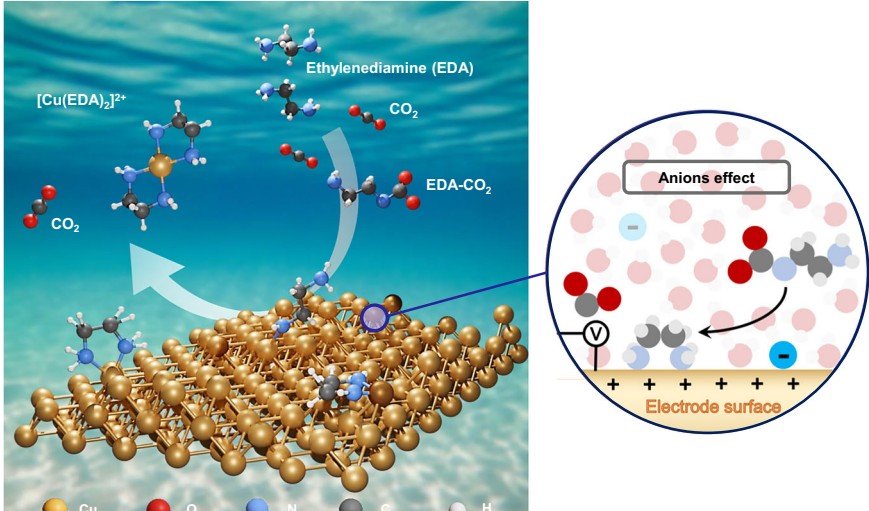

**Fig. 1 | The $CO_2$ release step as part of the EMAR process.** Illustration of the molecular interfacial processes during the electrochemical amine-mediated $CO_2$ release, the anodic partial process of electrochemically mediated amine regeneration (EMAR) systems. Shown is absorption $CO_2$ capture using ethylenediamine (EDA) to carbamate followed by the anodic electrochemical $Cu^{2+}$-mediated

competitive (indirect) $CO_2$ release from the carbamates (EDA-carbamates) generating the strong $[Cu(EDA)_2]^{2+}$ complex. Color coding is Red (O), gray (C), yellow (Cu), blue (N) and white (H). The inset at right illustrates that electrolyte anion adsorption (blue negative ball) affects the $CO_2$ capture and release kinetics by modulating the interacting of carbamate with the Cu electrode surface.

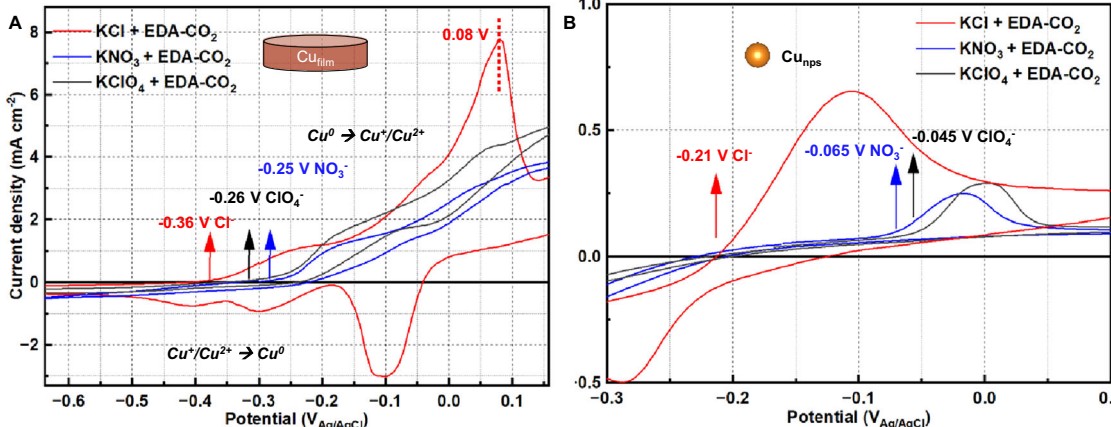

**Fig. 2 | The CO$_2$ release performance on Cu in H cell.** CO$_2$ release-cyclic voltammetry for **A** Cu foil and **B** Cu nanoparticles in 0.1 M KA (A = Cl$^-$, NO$_3^-$ and ClO$_4^-$) supporting electrolyte with 0.1 M EDA-CO$_2$ carbamates in H cell. All the CVs were recorded at 5 mV s$^{-1}$. The average resistance of Cu foil in 0.1 M EDA-CO$_2$ electrolyte containing 0.1 M KCl, KNO$_3$ and KClO$_4$ at 25 °C was 5.4, 5.5 and 5.4 Ω, respectively. For Cu nps, the corresponding values were 6.6, 6.5, and 6.4 Ω. All CV curves were recorded without iR correction. Arrows indicated the onset potentials for Cu oxidation extracted from the voltammograms, and the numerical voltage labels denoted the corresponding onset values (vs. Ag/AgCl) for each supporting electrolyte. The arrow at 0.08 V (vs. Ag/AgCl) showed a distinct oxidation peak uniquely observed in the KCl + EDA–CO$_2$ electrolyte.

computational methodological approaches highlight the importance of the local anion microenvironment for the design of EMAR systems, improving our understanding of the interfacial EMAR CO$_2$ release process.

## Results

### Voltammetry of CO$_2$ release on Cu in H-cell

To elucidate the role of electrolyte anions in the electrochemical CO$_2$ release step of the EMAR process, we started by investigating the cyclic voltammetry (CV) over Cu foil and Cu nanoparticle (NP) electrodes in various 0.1 M KA (A = Cl$^-$, NO$_3^-$, ClO$_4^-$, Br$^-$, and I$^-$) supporting electrolytes in a H-cell configuration in absence and presence of either 0.1 M ethylenediamine (EDA) or 0.1 M CO$_2$-saturated EDA carbamate (EDA-CO$_2$). Characteristic CV responses on the Cu electrodes for each anion and for all three electrolyte conditions were shown in Figs. S1 and S2. Anodic Cu/Cu$^{+/2+}$ redox waves emerged in the order of EDA, EDA-CO$_2$, and pure supporting electrolyte. EDA appeared to interact with the Cu surface very strongly and favored formation of Cu ions[27]. Fig. 2A and B evidenced that Cu foil and Cu NP in KCl + EDA-CO$_2$ system owned apparently faster onset kinetics among all the supporting electrolytes. Notably, Cu foil and Cu NPs displayed Cu oxidation onset potentials of −0.36 V$_{Ag/AgCl}$ and −0.21 V$_{Ag/AgCl}$ respectively in 0.1 M KCl + EDA-CO$_2$ electrolyte, which are significantly more cathodic compared to those obtained in KNO$_3$ + EDA-CO$_2$ and KClO$_4$ + EDA-CO$_2$ electrolyte. The faradaic efficiency (FE) for oxidation of the Cu foil to form [Cu(EDA)$_2$]$^{2+}$ (derived from quantification of the latter) was 90.2 ± 2.0% in 0.1 M KCl + EDA-CO$_2$ electrolyte (Fig. S3, see Method for details). Notably, cyclic voltammetry in Fig. 2A, conducted in 0.1 M KCl with EDA–CO$_2$ electrolyte, showed a distinct oxidation peak at -0.08 V vs. Ag/AgCl followed by a consistent sharp current drop, indicating that the electrode surface formed via a dissolution–precipitation mechanism, with the diffusing species likely being the CuCl$_2^-$ complex[28–31], whereas no obvious oxidative peak was observed in KNO$_3$ + EDA-CO$_2$ and KClO$_4$ + EDA-CO$_2$ electrolyte.

The more cathodic Cu oxidation wave in chloride-containing EDA-CO$_2$ suggests a much stronger interaction between Cl$^-$ anions and Cu$^{+/2+}$[32,33]. Indeed, chloride anions are known to readily undergo complexation with copper cations at or near the surface, specifically Cu ions. This Cu-Cl complexation likely acts as a thermodynamic driving force favoring Cu oxidation, thus shifting the Nernstian Cu redox potential, in turn shifting the apparent kinetic onset potential. Once interacting with Cl$^-$ anions, Cu$^{2+}$ ions were generated and subsequently complexed with EDA according to Eq. (3), inducing the release of free CO$_2$ into the solution. In addition, the anion concentration exhibited only a minimal impact on CO$_2$ release behavior (Fig. S4). As shown in Fig. S5, similar Cl$^-$-promoted Cu$^{2+}$ release and CO$_2$ desorption were also observed using N-methyldiethanolamine (MDEA), a tertiary amine with weaker Cu$^{2+}$ binding affinity. This suggests that the anion effect could potentially extend to other classes of amines with varying coordination strengths and structures.

### Real-time kinetic study of interfacial CO$_2$ release on Cu

To evaluate the precise kinetic onset potential of the EMAR CO$_2$ release, time-resolved detection of evolved product gases at the interface as a function of applied potential was conducted using DEMS, which allowed the simultaneous recording of voltametric responses and product-specific ion current signals[34]. DEMS measurements were carried out in saturated EDA-CO$_2$ carbamate electrolyte in a custom-made capillary flow cell in presence of three different KA electrolytes (A= Cl$^-$, NO$_3^-$ and ClO$_4^-$), and the applied electrode potential on a Cu disc electrode was cycled ~10 times until a stable CV had been reached. The capillary flow cell showed in Fig. S6 combines a continuous convective electrolyte flow toward the electrified interface with a rapid continuous capillary sampling of products toward the liquid–vacuum interface. This ensured plug-flow conditions and sub-second response times for all volatile electrochemical products. During the anodic sweep, the CO$_2$ evolution onset potential was defined as the potential at which the mass spectrometer ion current signal for the m/z = 44 species (CO$_2$) reached 2% of the peak value observed during CV cycling, a reasonably low detection threshold to consider as the CO$_2$ release onset. The DEMS results (Fig. 3) revealed the detailed potential dependence of CO$_2$ evolution, with interfacial molecular CO$_2$ intensity increasing monotonically during anodic potential sweeps, consistent with anodic CO$_2$ release at the Cu surface according to Eqs. (2) and (3). No competing O$_2$ evolution (Fig. S7) was detected during anodic polarization, confirming a high faradaic efficiency of the CO$_2$ release process. The limited faradaic losses in Fig. S3 may be related to competing processes near the double layer, which likely include the formation of free Cu$^{2+}$ and Cu−Cl species. While presence of free EDA in the feed would also result in inefficiencies due to [Cu(EDA)$_2$]$^{2+}$ complex formation, this inefficiency route is likely not dominant under our CO$_2$ saturated carbamate feed conditions. Experimental kinetic onset potential values of CO$_2$ release showed a reproducible trend with the nature of the anion, in particular in the order Cl$^-$ < NO$_3^-$ < ClO$_4^-$. More

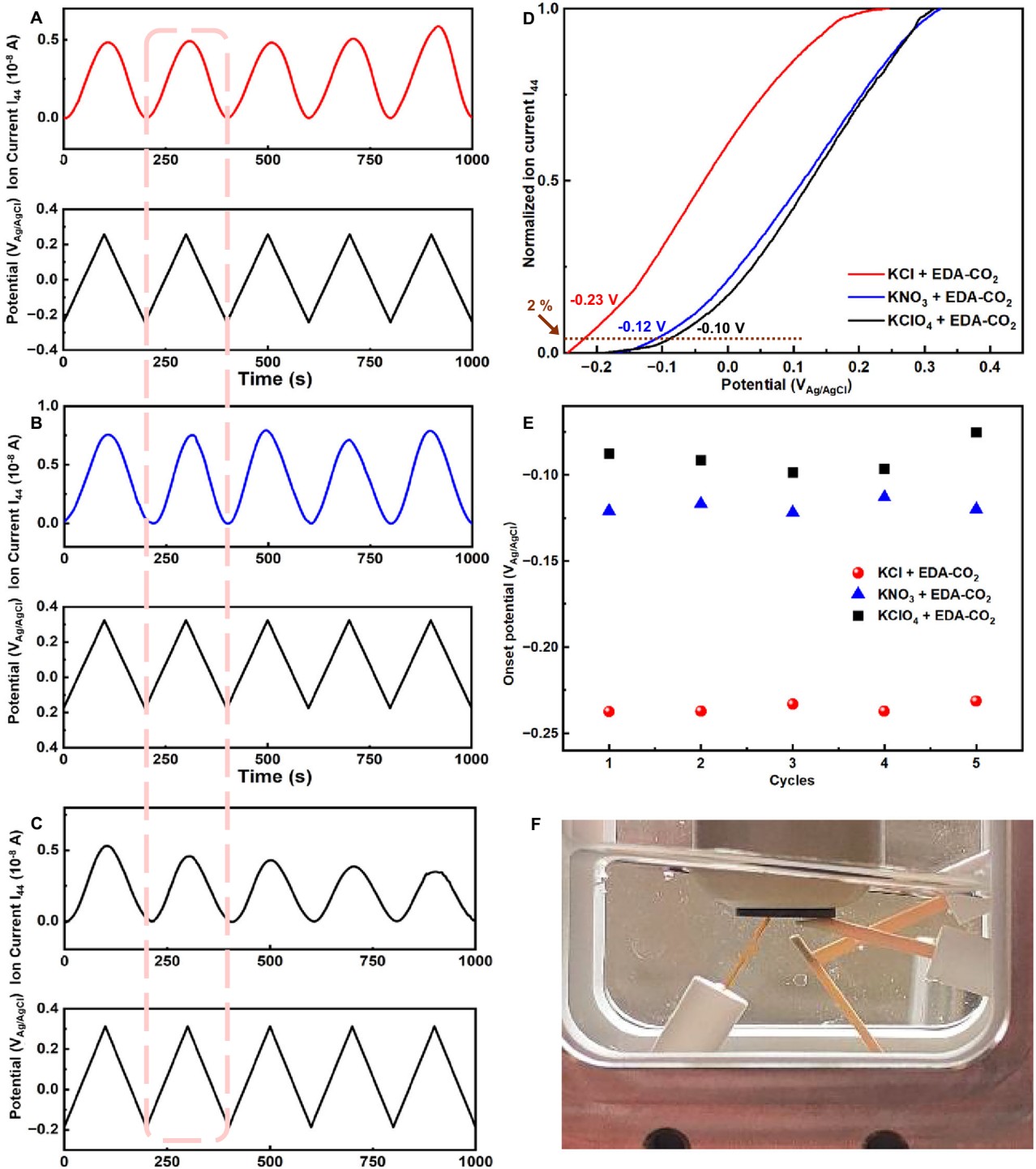

**Fig. 3 | Differential Electrochemical Mass Spectrometry (DEMS) studies of the anion impact on EMAR interfacial $CO_2$ release kinetics.** Real-time DEMS analysis links the applied Cu electrode voltametric potential sweeps to the m/z = 44 mass current signal revealing kinetic onsets of interfacial $CO_2$ release in 0.1 M EDA-$CO_2$ (carbamate) electrolyte by continuous cyclic voltammetric scan at 5 mV s$^{-1}$ in presence of 0.1 M **A** KCl, **B** KNO$_3$ and **C** KClO$_4$. The bottom plot shows the concurrent cyclic voltametric sweep over time. **D** Voltammetric potential sweeps with the m/z = 44 mass current signal corresponding to the cycle outlined by dashed lines in (**A**–**C**). Arrows and numerical labels marked the $CO_2$ release onset potentials versus Ag/AgCl, defined as the anodic potential at which the m/z = 44 ion current reaches

2% of its peak value. The numerical voltages indicated the corresponding onset potential for each supporting electrolyte. **E** Comparison of the $CO_2$ evolution onset potentials for each cycle in various supporting electrolytes, as detected by an operando DEMS capillary flow cell system. The kinetic onset potential was defined as the potential at which the mass spectrum ion current signal reached 2% of peak maximum for the m/z = 44 species. CVs were recorded by applying potential cycles within a specific range. The potential windows for each electrolyte were selected relative to their respective open-circuit potentials (OCP). The scan begins at the open-circuit potential, where no $CO_2$ release occurs. Electrolyte flow is 2 µL s$^{-1}$. **F** Photo of custom-made capillary flow cell.

specifically, Cu disc electrodes displayed a kinetic $CO_2$ onset potential of $-0.23$ $V_{Ag/AgCl}$ in KCl (Fig. 3D and E), $-0.12$ $V_{Ag/AgCl}$ in $KNO_3$, and $-0.10$ $V_{Ag/AgCl}$ in $KClO_4$. We conclude that our DEMS investigations firstly provided accurate onset potentials for EMAR-based $CO_2$ release. Different supporting electrolytes revealed a significant anion impact at the Cu-EMAR interface, and thus suggesting the presence of $Cl^-$ induces earlier $CO_2$ release.

Since the applied overpotential for an electrochemical reaction is related to the amount of excess energy needed to drive the reaction, we sought to estimate and compare values for this "excess energy consumption" of the EMAR anode process operating in the different electrolytes. The calculation, using our measured values of $CO_2$ onset potential (Fig. 3D and Table S1) related to the measured open circuit potential (Fig. S8), and the measured faradaic efficiency (Fig. S3), is explained in Supplementary Note 2. By this approach, we estimate that initiating $CO_2$ release in the KCl + EDA-$CO_2$ system required an excess energy consumption of 0.64 $kJ_e/mol_{CO_2}$ to overcome overpotential barriers, which is significantly lower than the energy requirements for the $KNO_3$ + EDA-$CO_2$ (6.4 $kJ_e/mol_{CO_2}$) and $KClO_4$ + EDA-$CO_2$ (10.2 $kJ_e/mol_{CO_2}$) systems, corresponding to energy savings of 5.76 $kJ_e/mol_{CO_2}$ compared to the $KNO_3$ + EDA-$CO_2$ system and 9.56 $kJ_e/mol_{CO_2}$ compared to $KClO_4$ + EDA-$CO_2$ system. These results highlight the importance of optimizing the electrolyte composition to minimize energy consumption in electrochemical $CO_2$ capture and release processes. Comparison of our system with the state-of-the-art in regeneration energy consumption is inherently challenging due to the significant variations in assumptions across the studies reviewed. This complexity is exemplified by a statement regarding the relatively mature Carbon Engineering Direct Air Capture (DAC) and Negative Emissions Technologies (NETs) process: "CE has spent several tens of millions of dollars developing DAC technology, yet our performance and cost estimates still carry substantial uncertainty."[35] Thus, to ensure a focused comparison on the energetics of a specific half-cell electrode process, we consider only the core components of the electrochemical system related to overpotential.

### Real-time tracking of interfacial Cu-amine complexation

In situ UV-vis absorption spectroscopy experiments were conducted to track the evolution of electrochemically generated Cu complex species at the surface diffusion layer according to Eq. (3), wherein the generation of colored $Cu(Am)_n^{2+}$ can be detected by absorption spectroscopy (Fig. S9)[36]. The UV-vis spectroelectrochemical cell as shown in Fig. S10 includes a patterned "honeycomb" electrode mounted in a thin-layer quartz cuvette, with a secure cap ensuring precise alignment of the electrode and reference. UV-vis spectra were collected in 0.1 M KCl + EDA-$CO_2$ electrolyte under potential sweeps from $-0.24$ to $+0.36$ $V_{Ag/AgCl}$, which corresponds to 0–0.6 $V_{OCP}$. At open circuit potentials (OCP) of $\sim-0.24$ $V_{Ag/AgCl}$, there was no observable absorption in the 400–800 nm wavelength range, confirming no Cu dissolution before EMAR $CO_2$ release reaction. With increasing applied anodic potential from $-0.24$ to $+0.36$ $V_{Ag/AgCl}$, absorption at 550 nm gradually increased (Fig. 4A and B), which was attributed to the formation of the colored $[Cu(EDA)_2]^{2+}$ complex by the re-complexation reaction of Eq. (3). As the electrode potential decreased, the 550 nm peak intensity slowly faded due to diffusion (Fig. 4B). To compare anion effects, the measurements were also carried out in $KNO_3$ and $KClO_4$ electrolyte. They displayed similar spectroscopic behavior (Fig. S11). It is noted that after three cycles of EMAR $CO_2$ release reaction, only the voltametric sweep in KCl showed sustained $[Cu(EDA)]^{2+}$ absorbance Fig. S12, given Fig. S11 as a reference, confirming a fast formation of $[Cu(EDA)_2]^{2+}$, and thus $CO_2$ released faster in presence of KCl than in other KA electrolytes.

We further study the EMAR $CO_2$ release kinetics of Eq. (3) using in-situ Fourier Transform Infrared Spectroscopy (FT-IR) in an ATR configuration. Thereby we probed surface reaction intermediates and

product over Cu NP electrodes[37,38]. No free $CO_2$ absorption could be observed at open circuit potential of $-0.24$ $V_{Ag/AgCl}$. FTIR spectra recorded during anodic voltammetric scans from OCP ($-0.24$ V to $+0.26$ $V_{Ag/AgCl}$, corresponding to 0–0.5 $V_{OCP}$) revealed a distinct vibrational feature at 2340 $cm^{-1}$ (Fig. 4C, D, and S13), attributable to linear free $CO_2$, confirming its evolution according to Eq. (3). On the cathodic scan ($+0.26$ $V_{Ag/AgCl}$ to OCP), the intensity of $-2340$ $cm^{-1}$ gradually decreased and eventually disappeared, in accordance with the decreasing reaction rate.

Our experimental in situ studies on the EMAR $CO_2$ release process of Eq. (3) revealed consistency between in situ FTIR, in-situ UV-vis and real-time DEMS data. The above in situ spectroelectrochemical results presented clarified that $Cl^-$ effect plays a crucial role in interfacial interactions, influencing the formation of Cu-amine complexation step and $CO_2$ release step[25].

### Interfacial molecular dynamics simulations

To elucidate the role of anions in modulating the onset potential kinetics of $CO_2$ release, molecular dynamics (MD) simulations on the Cu/electrolytes interfaces were performed for the investigated anions of chloride, nitrate and perchlorate. Our computations could help assess how variations in electrolyte chemistry could affect EMAR $CO_2$ release performance, primarily through modifications in the electrochemical double-layer structure[39,40]. The systems were constructed following established methodologies in the literature, where the liquid electrolytes involving the associated ions and molecules were placed between two Cu electrodes[39,40]. Assuming that the local interfacial accumulation of carbamates and $CO_2$ molecules around the active electrode will promote the subsequent $CO_2$ release, we investigated the anion effect through MD simulations, specially qualifying and comparing the distribution behavior of anions and these two molecules. As shown in Fig. 5A–C, at the constant potential difference of 0 V (no potential difference), a considerable accumulation tendency of $Cl^-$ ions was observed around the bottom Cu electrode (anode), while contrastingly, nitrate and perchlorate anions distributed more uniformly throughout the electrolyte. This preferential local enrichment of $Cl^-$ at the Cu electrode is evident from the density peak (Fig. 5D), suggesting the distinct electrostatic adsorption between anions and the Cu surface. When analyzing the origin of the anion interaction discrepancy, the intrinsic features of anions such as size configurations, ionic conductivity and mobility, the diffusion coefficient may collectively make contributions, leading to the distinct ion distribution[40,41]. Variations in the specific anion adsorption control the local electric fields near the Cu electrode surface (extent of diffuse double layer for instance), thereby mediating the distribution of the molecular compounds[39,40]. As evidenced in Fig. 5A–C, the near-surface density of EDA-$CO_2$ carbamate and $CO_2$ molecules in the case of chloride anions was also higher than that in nitrate and perchlorate anions, for which the neutral molecules appeared more dispersed in the electrolyte. This enhanced accumulation of molecules could be intuitively observed from the density peak near the Cu surface, demonstrating the role of $Cl^-$ anions in promoting molecular adsorption (Fig. 5E and F). As a result, $Cu^{2+}$ will be generated more readily, potentially promoting an earlier release of molecular $CO_2$ from carbamate.

In addition, the application of an external cell voltage appears to further stimulate the inhomogeneous distribution of ions and molecules at the Cu electrode surface, accelerating the anion-induced Cu ion formation and subsequent $CO_2$ release process. As reflected in Fig. S14, at a constant cell voltage of 0.1 V (the potential of bottom anode is 0.1 V higher than that of the top cathode), more $Cl^-$ ions were found to aggregate near the anodic Cu surface. While the near surface concentrations of $ClO_4^-$ and $NO_3^-$ rose somewhat, their distribution remained spatially more uniform compared to $Cl^-$. The local accumulation of molecular compounds ($CO_2$ and Carbamate) near the Cu

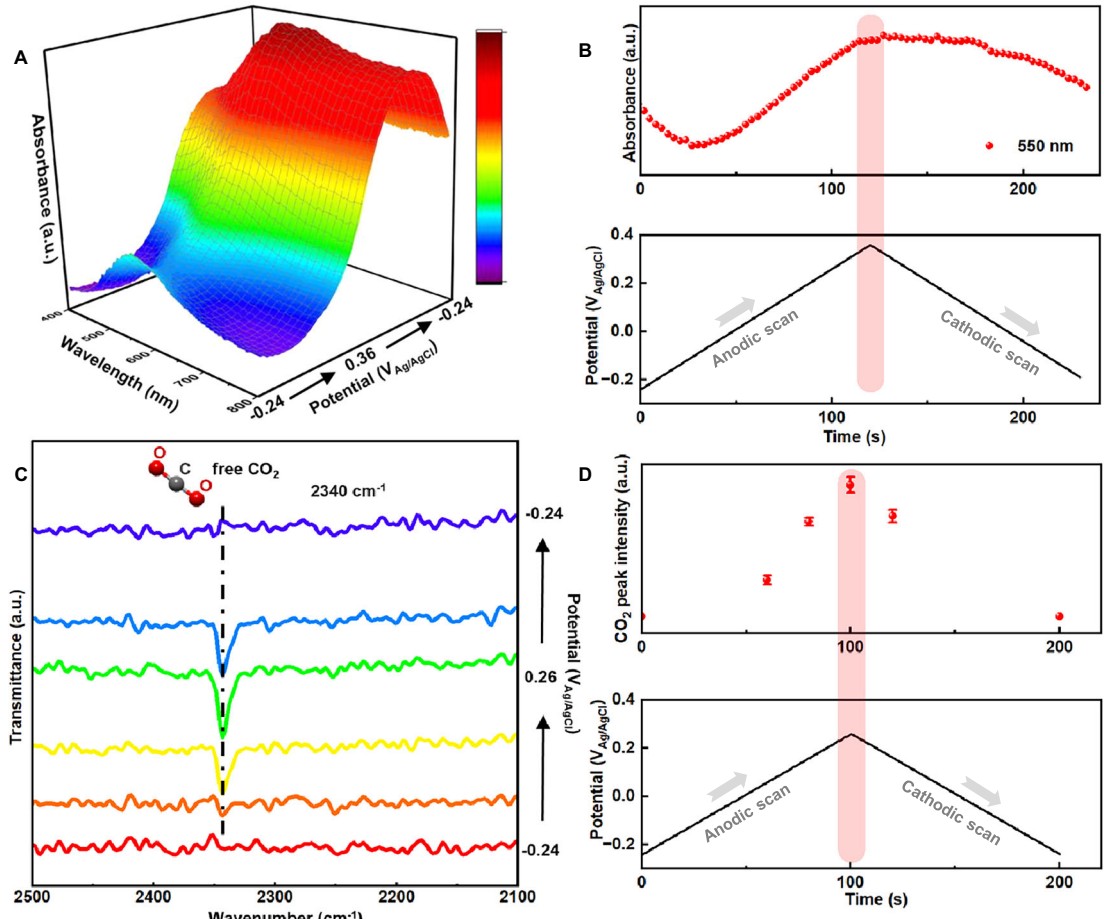

**Fig. 4 | In situ studies on $CO_2$ release at the Cu-EMAR interface. A** In situ UV-vis revealing $CO_2$ release on Cu nanoparticles in 0.1 M KCl scanned from −0.24 to +0.36 $V_{Ag/AgCl}$, **B** normalized peak area of Cu-amine complex signals obtained from in situ UV-vis in (**A**). The peak centered at 550 nm can be attributed to $[Cu(EDA)_2]^{2+}$. The bottom plot shows the concurrent cyclic voltammetric sweep over time. Spectra were collected for wavelengths between 400 and 800 nm with a time resolution of 400 ms at a scan rate of 5 mV s⁻¹. **C** In situ FTIR revealing $CO_2$ release on Cu nanoparticles as a function of applied potential in 0.1 M KCl, **D** normalized peak area of $CO_2$ release signals on Cu nanoparticles obtained from in situ FTIR in (**C**). The peak at 2340 cm⁻¹ could be assigned to released $CO_2$. The bottom plot shows the concurrent cyclic voltammetric sweep over time. All the FTIR spectra were obtained in the range of 2500 and 2100 cm⁻¹ with a spectral resolution of 4 cm⁻¹. Each spectrum was collected from an average of 64 scans. The potential was scanned at a scan rate of 5 mV s⁻¹. The background spectrum was taken before applying any potential. Error bars represent the ± s. d. between triplicate measurements.

anode increased, as well, in line with variations of the anionic near-surface charge and the corresponding local electric field at the Cu surface under non-equilibrium conditions. Strong Cu-Cl interaction with chloride anions could facilitate the oxidation of Cu to $Cu^{2+}$ ion, and thus enhance the $CO_2$ release dynamics (Fig. S15)[28,29,32,33,42]. Thus, the choice of the anions at the interfacial microenvironment offers an effective strategy to modulate the $CO_2$ release kinetics.

## Discussion

We have investigated the interfacial $CO_2$ release step of the EMAR process using in situ spectroscopy (FTIR, UV-vis) and real time DEMS. The $CO_2$ release step is a complex reaction step coupling interfacial anodic $Cu^{2+}$ ion release with re-complexation yielding free $CO_2$ molecules (Eqs. 1–3). In our investigations, scientific focus was placed on the impact of the nature of electrolyte anions. We have demonstrated a $Cl^-$-mediated electrochemical anodic $CO_2$ release with a significantly lower (less positive) and hence energetically more favorable onset potential compared to other electrolyte anions. The $Cl^-$ ions enabled onset potentials as low as −0.36 $V_{Ag/AgCl}$ on Cu electrodes. Trends in onset potentials were confirmed for Cu nanoparticles and foils. Poor activity was observed in KI and KBr electrolytes due to the formation of CuA precipitates. In situ FTIR and in situ UV-vis analyses confirmed the

formation of $CO_2$ and $[Cu(EDA)_2]^{2+}$, with their concentrations increasing and decreasing in response to the applied potential. UV-vis data enabled the estimation of faradaic efficiencies of over 90% for the EMAR $CO_2$ release process for each anion. Parasitic residual amine Cu corrosion and formation of $Cu^+$ surface species can account for the imperfect faradic efficiencies. Finally, MD calculations suggested anion-regulated preferentially distribution of neutral $CO_2$ and carbamates molecules near the Cu interface in the Cu-Cl system favoring the ready formation of $Cu^{2+}$ ions associated with lower $CO_2$ onset potentials.

To our knowledge, this is the first in-situ, time-resolved study of the anodic interfacial molecular $CO_2$ release in the EMAR desorption and regeneration process. We expect that this report will spark wider use of interfacial in-situ analytics to provide insight into the kinetic barriers and mechanisms of molecular interfacial processes on solid electrodes during the electrified $CO_2$ capture, release, and concentration. Besides, within the KA + EDA-$CO_2$ system, our findings reveal that initiating $CO_2$ release in the KCl + EDA- $CO_2$ configuration requires an excess energy consumption of 0.64 kJ$_e$/mol$_{CO2}$— substantially lower than the 6.4 kJ$_e$/mol$_{CO2}$ and 10.2 kJ$_e$/mol$_{CO2}$ required by the KNO$_3$ + EDA-$CO_2$ and KClO$_4$ + EDA-$CO_2$ systems, respectively—thereby achieving energy savings of 5.76 kJ$_e$/mol$_{CO2}$ and 9.56 kJ$_e$/mol$_{CO2}$.

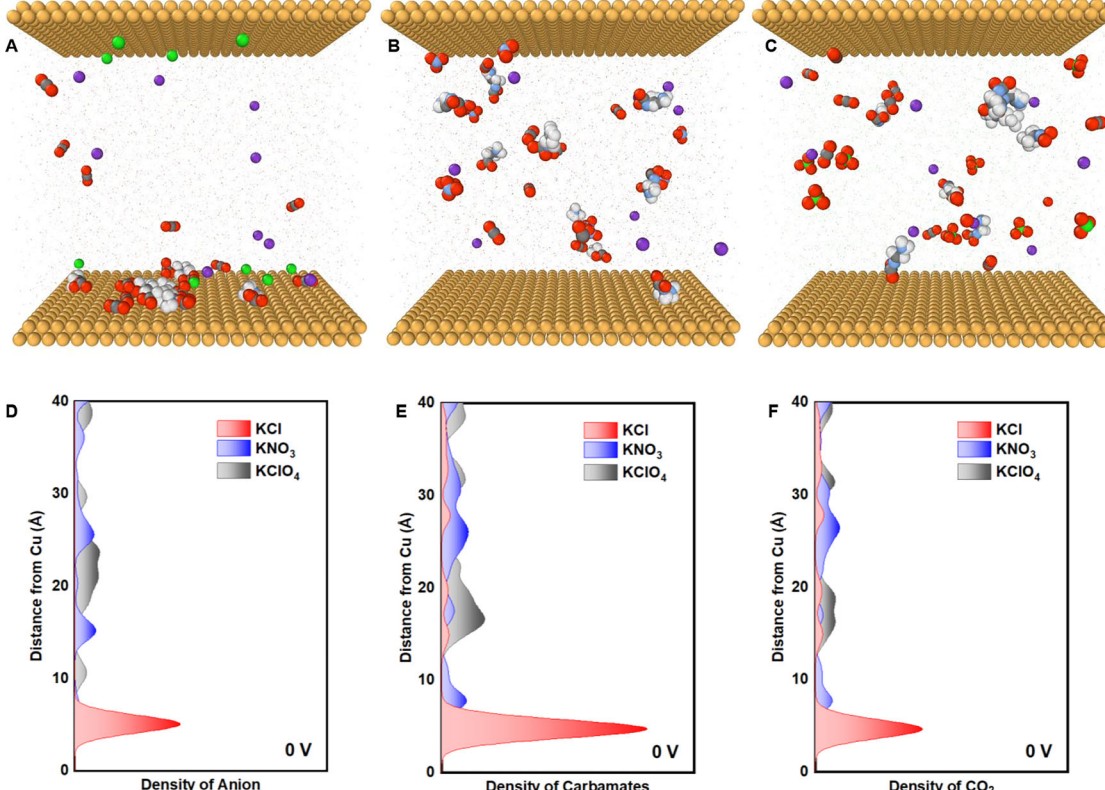

**Fig. 5 | Computational studies of the anion impact on EMAR interfacial CO₂ release dynamics.** Molecular dynamics predictions of atomic configurations for the **A** Cu-carbamates-Cl interface, **B** Cu-carbamates-nitrate interface and **C** Cu-carbamates-perchlorate interface at the fixed potential difference (ΔU) of 0 V. Comparison of density distribution of **D** anions, **E** carbamates and **F** CO₂ molecules versus distance from the Cu electrode. ΔU, voltage difference between the bottom anode and top cathode of the Cu(111) surfaces, ΔU = 0 V indicates no voltage difference. The concentrations of molecules and anions were set to approximately 0.1 M to align with the experimental conditions. Color coding is Red (O), gray (C), yellow (Cu), green (Cl), purple (K), blue (N) and white (H).

MD calculations further provided theoretical insights into the anion-regulated preferential distribution of neutral CO₂ and carbamates molecules at the electrode interface within the Cu-Cl system, thereby favoring the formation of Cu²⁺ ions associated with lower CO₂ onset potentials. Cl⁻ ions tended to accumulate around the Cu electrode, leading to a preferential distribution of neutral CO₂ and carbamates molecules. In contrast, NO₃⁻ and ClO₄⁻ show inferior affinity to the Cu electrode, inducing the suboptimal interfacial accumulation of neutral molecules. This heterogeneous molecular distribution implies the anion-modulated molecular interactions at the Cu surface, wherein specific ionic solvation dynamically controls the local electrical microenvironment, thereby dictating the spatial arrangement of adsorbates. Our findings on the anion effect offer a facile strategy to lower regeneration energy consumption in NETs and DAC, while our insights into electrolyte compatibility and interfacial engineering enable localized, controllable CO₂ release, providing a foundation for integration with downstream CO₂ conversion.

## Methods
### Materials
Potassium chloride (KCl, ≥99%) and potassium nitrate (KNO₃, ≥99%) were purchased from VWR. Potassium perchlorate (KClO4, ≥99%) was purchased from Thermoscientific. Copper nanoparticles (nanopowder, 25 nm particle size) and ethylenediamine (≥99%) were purchased from Sigma Co., Ltd.

### Preparation of Cu nanoparticle electrode
50 μL of commercial Cu nanoparticle ink (0.125 mg cm⁻² loading) was drop-cast onto the polished glassy carbon working electrode. The ink was prepared by dispersing 20 mg of powder in 8 mL of water, 1.5 mL of isopropanol, and 50 μL of a 5 wt% Nafion solution, followed by sonication.

### Preparation of Cu foil electrode
**Electrodes polishing.** Before use, copper foil (99.99%, 1 mm thickness, 1 cm × 2.5 cm) was polished via electropolishing in 85% phosphoric acid by applying a potential of + 1 V for 1 min. Both sides of copper foil were exposed to solution. Copper electrode, Pt mesh and Ag/AgCl was used as the working electrode, counter electrode and reference electrode, respectively.

### Characterizations
**Electrochemical measurements.** The electrochemical tests were carried out in H cell, separated by a Nafion 117 membrane (Ion Power, GmbH). Prior to use, the membrane was sequentially treated with H₂O₂, H₂SO₄ and deionized water at 80 °C. The experiments were conducted in freshly prepared aqueous solutions containing 0.1 M ethylenediamine with different supporting electrolytes (KCl, KNO₃, KClO₄, KI, and KBr). Cu electrodes, platinum mesh, and a leak-free Ag/AgCl electrode were used as working electrodes, counter electrode, and reference electrode, respectively, measured with a Biologic SP 300 potentiostat. The leak-free Ag/AgCl electrode was calibrated versus a Pt/H₂ electrode. Before the electrochemical reaction, electrolyte in the working compartment was sparged continuously with CO₂ overnight at a flow rate of 10 sccm (standard cubic centimeters per minute) from the bottom of the cell, triggering the formation of Am-CO₂ (Eq. 1). The pH of CO₂-saturated 0.1 M EDA−CO₂ electrolyte containing 0.1 M KCl, KNO₃ and KClO₄ at 25 °C (7.2 ± 0.02, 7.1 ± 0.02, and 7.2 ± 0.02,

respectively) was measured using a calibrated pH meter. All CV curves recorded in the H-cell were presented without iR correction. The resistance was measured using the electrochemical workstation prior to the measurements. For each measurement, fresh electrolyte was used.

In situ **FTIR spectroscopy** was carried out in a modified attenuated total reflection (ATR)-FTIR setup. The Cu nanoparticles were dispersed in water, and the solution of samples was deposited directly onto the ATR prism, avoiding intensity loss due to the lack of a sputtered layer of a conducting material such as Au. The electrochemical connection was made via a carbon cloth material that provides a reservoir of electrolyte over the samples, minimizing mass transport limitations and polarization effects as shown in Fig. S16. The spectra were collected in a custom-made glass cell with a Bruker Vertex 70 v FTIR spectrometer equipped with a Mercury-Cadmium-Telluride detector cooled with liquid nitrogen. A platinum mesh and a leak-free Ag/AgCl were used as counter electrode and reference electrode, respectively. All in situ electrochemical measurements were controlled using a Metrohm Autolab PGSTAT204 potentiostat. An unpolarized beam was focused with a Pike Veemax II onto the sample spot of the cell. The spectral resolution was set to $4\,cm^{-1}$ and 100 to 107 interferograms were collected and averaged for each presented spectrum. Here, the reference spectrum was collected in the same electrolyte (0.1 M $CO_2$ saturated EDA with various supporting electrolytes) immediately before the investigated potential scan at the respective start potential. A Si hemisphere was used as the ATR crystal and sample was deposited on the prism on the IR beam ATR focus spot and contacted with Toray Paper 030 carbon cloth and a Pine glassy carbon rod fixating the carbon cloth. The complete beam pathway was under vacuum more than 24 h prior to each measurement. The positions of the bands were determined by taking the band maximum, or, for bipolar bands, by taking the mean value of the maximum and the minimum. The potential was scanned over the specified range at a scan rate of $5\,mV\,s^{-1}$. The background spectrum was taken before applying any potential.

In situ **UV−vis spectra** were collected using a UV/VIS/NIR spectrometer from Avantes (AvaLight-DH-S-BAL) equipped with a deuterium and halogen lamp and a light guide. The spectrometer was connected via fiber optic cables to the electrochemical cell. Transmittance spectra were collected between 400 and 800 nm with a time resolution of 400 ms. The electrochemical cell constituted a 1 cm quartz cuvette, Ag/AgCl reference electrode with a Ø = 1 mm shaft (Pine ltd.). The potential was scanned over the specified range at a scan rate of $5\,mV\,s^{-1}$, measured with a Biologic SP 300 potentiostat. The background spectrum was taken before applying any potential.

DEMS was measured in a dual thin layer electrochemical flow cell from Liquidloop ltd., "PoGASi DEMS 3". DEMS was recorded in 0.1 M $CO_2$ saturated EDA with various supporting electrolytes. The electrochemistry was controlled using a BioLogic potentiostat, a leak-free Ag/AgCl as a reference electrode (Warner Instruments), and a Pt mesh as a counter electrode. The DEMS cell was connected to a mass spectrometer (QMS 200, Pfeiffer Vacuum) via a 50 μm thick microporous PTFE membrane with 20 nm pore size (Cobetter®, Cat. No. PF-002HS). Two turbomolecular pumps (HiPace 80) were operating at $10^{-6}$ mbar. Electropolished Cu disc electrodes were used as working electrode, CVs were recorded by potential cycling at a specific range, measured by a Biologic SP 300 potentiostat. The scan limits varied slightly for the different electrolyte pHs to account for differences in resistance and activity. The samples were first preconditioned at a scan rate of $5\,mV\,s^{-1}$ for -10 cycles.

### Classical molecular dynamics (MD) simulations
To investigate the distribution of $CO_2$ and $CO_2$-EDA around Cu electrode surface in the presence of different anions (chloride, nitrate, and perchlorate), molecular dynamics (MD) simulations were performed to examine the confined behavior of anions, $CO_2$, and $CO_2$-EDA in aqueous electrolyte system between two Cu electrodes. The simulation system was constructed using PACKMOL[43], with a box size of 62.6 Å × 41.0 Å × 70 Å, the atomic positions of the Cu(111) electrodes were frozen during the entire simulation. Periodic boundary conditions (PBC) were applied in the x and y directions, while the z direction was fixed. The system contained 4560 water molecules, 9 EDA-$CO_2$ carbamate and $CO_2$ molecules, 9 anions. $K^+$ ions were introduced to ensure charge neutrality, corresponding to an anion concentration of approximately 0.1 M. Short-range interactions between molecules were described by Lennard-Jones (LJ) potentials, while long-range Coulombic interactions were calculated using the particle-particle particle-mesh (PPPM) method, with a precision of $10^{-5}$ and a cutoff radius of 12 Å.

The force field parameters for the system were obtained from multiple references: Cu atoms from ref. 44, $Cl^-$ from ref. 41, $NO_3^-$ from ref. 42, $ClO_4^-$ from ref. 45, $K^+$ from ref. 46, and $CO_2$ from ref. 47. The parameters for $CO_2$-EDA were based on the second-generation general Amber force field (GAFF2)[48] and generated using the AuToFF[49] tool. Cross-interaction parameters between atoms were calculated using the Lorentz-Berthelot mixing rules[50]. To eliminate initial energy hotspots, energy minimization was performed before each simulation, followed by a 2 ns preliminary equilibration simulation in the NVE ensemble. This ensured the establishment of a thermodynamically stable interfacial structure for subsequent simulations. In the production simulation phase, the equilibrated system was transferred to a 2D-PBC setting, and electrode potentials were adjusted using the USER-CONP2 package[51], which implements the finite field method. Simulations were performed at a fixed potential difference of ΔU = 0.0 V and 0.1 V in the NVT ensemble at 298.15 K for at least 10 ns. Temperature control was achieved using a Nosé-Hoover thermostat, with Hamiltonian dynamics regulated by global velocity rescaling[52]. The final 1 ns of trajectory data was analyzed to ensure statistical reliability. Data visualization was performed using OVITO software to reveal key features of the interfacial microstructure and molecule affinity behavior[53].

### Quantitative analysis by UV-vis absorption spectroscopy of [Cu(EDA)₂]²⁺ complex
The $[Cu(EDA)_2]^{2+}$ complex was analyzed by UV-vis spectroscopy using a UV/VIS/NIR spectrometer from Avantes (AvaLight-DH-S-BAL) equipped with a deuterium and halogen lamp and a light guide. The Cu-EDA complex exhibits an absorbance at 550 nm, so the selected wavelength range was 400−800 nm. A blank solution containing 0.1 M EDA and KCl was used for baseline correction by filling a cuvette, inserting it into the spectrophotometer, and setting the absorbance to zero. A calibration curve was prepared using standard solutions with known concentrations (1.64, 1.75, 1.85, 1.93 and 2.00 mM), measuring their absorbance at 550 nm, and plotting absorbance against concentration. The sample absorbance was measured at 550 nm, with samples collected at various potentials after 30 min of reaction. The sample concentration was determined using the calibration curve with the formula: $c = A - y_{intercept}/slope$. The calibration curve for UV-vis spectrophotometric determination of $[Cu(EDA)_2]^{2+}$ complex was shown in Fig. S17.

Faradaic efficiency, FE, was expressed as the ratio of the actual charge involving Cu redox reaction to its theoretical charge, is calculated by dividing the observed concentration of Cu-EDA complex.

$$FE = \frac{2n_{Cu}F}{\int_0^{t_0} I dt}$$

Where $n_{Cu}$ are the moles of Cu, determined by measuring the concentration of the Cu-amine complex using UV-vis spectroscopy.

## Data availability

The MD simulation structures are provided at https://doi.org/10.5281/zenodo.17501792. Source data are provided with this paper.

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

## Acknowledgements

This work was financially supported by the Sol2H2 project of European Union's Horizon 2020 research and innovation program under grant No. 101006701, EcoFuel, the Deutsche Forschungsgemeinschaft (DFG, German Research Foundation, grant No. STR 596/18-1 and LI 4184/2-1), Natural Science Foundation of Anhui Province (2308085MB47), Anhui Provincial Department of Education university natural science research project (2024AH050070), and the Initiative and Networking Fund of the Helmholtz Association (grant agreement No. KA2-HSC-12, 'A Comprehensive Approach to Harnessing the Innovation Potential of Direct Air Capture and Storage for Reaching $CO_2$-Neutrality', DACStorE). The authors thank Hefei Advanced Computing Center and the High-Performance Computing Platform of Anhui University for computational support.

## Author contributions

P.S. and L.L. conceived the idea and co-wrote the paper; L.L. carried out the electrode preparation, characterization, and electrochemical evaluation; F.F. conducted the DEMS measurements and analyzed data; L.Y. and J.W. carried out the MD simulation and analyzed the results; L.Y. supervised the MD simulation; X.W. and W.J. participated in the discussion of the electrochemical results; M.M. participated in the discussion of reaction mechanism; all authors contributed to the discussions and assisted during manuscript preparation.

## Funding

## Competing interests

The authors declare no competing interests.
