## [Transparent Peer Review file · Nature Communications]

Anion Effects Govern Efficiency of Electrochemical Amine-Mediated CO₂ Capture/Release

Corresponding Author: Professor Peter Strasser

Version 0:

Reviewer comments:

Reviewer #1

(Remarks to the Author)

Reviewer #2

(Remarks to the Author)

The authors proposed a Cl-mediated EMAR CO₂ release pathway, which reported a lower CO₂ release onset potential. Time-resolved DEMS, in situ UV-vis, in situ FTIR, and MD calculations were combined to provide relatively complete evidence to clarify the importance of local anion microenvironment in EMAR systems. The research focus of this work is interesting and meaningful in EMAR research, while the data analysis should be strengthened to make the conclusion more persuasive.

1. Could the authors provide kinetics or thermodynamics data about the complex reaction between Cu⁺² and Cl⁻, and compare it with that of Cu⁺² and EDA? As shown in Equation (3), the formation of Cu-Am complex is already a very thermodynamically preferred reaction. If the coordination stability of Cu⁺² and Cl⁻ is stronger, would the formation of Cu-Cl complex hinder the release of CO₂?
2. There was an oxidation peak at 0.08 V_{Ag/AgCl} only in KCl + EDA-CO₂ electrolyte as shown in Fig. 2A. What is the physics chemical process corresponding?
3. The authors attributed the loss of FE in Fig. S3 to direct Cu oxidation and [Cu(EDA)₂]²⁺, which is confusing. According to Methods, FE means the ratio of charge used to form [Cu(EDA)₂]²⁺ and total charge, which seems to imply the lost charge was used to form Cu₂⁺, Cu-Cl complex, or Cl₂, as no O₂ evolution was detected.
4. The authors declared their Cl-mediated interfacial EMAR CO₂ release pathway enables “one of the lowest ever reported onset potentials of CO₂ release”, which should be supported by data from other works.
5. Please give a clear statement of potential ranges in captions of Figs. 3A-C. What is the reason to choose different potential ranges for three electrolytes? Would a more negative potential range used in KCl + EDA-CO₂ cause a more negative onset potential?
6. The authors attributed the decreasing FTIR intensity of free CO₂ during the cathodic scan to the decreasing reaction rate. Why is the signal of CO₂ not continuously accumulating?
7. The discussion about the effects of baseline or data in the background was missing in several experiments:
 - a. The Am-CO₂ electrolyte was obtained after CO₂ flow overnight at a flow rate of 10 sccm, which is likely to form a CO₂-over-saturated solution. Would it lead to a continuous CO₂ release without any electrochemical experiments? Should this spontaneous CO₂ release be excluded as a baseline when analyzing data?
 - b. In Figs. 3A-3C, three current signals showed different trends: the current signal in 3A is very stable, it in 3B starts almost from 0, while it in 3C shows a decreasing trend. Would it be related to the baseline in the measurements?
 - c. In Fig. S10, it is suggested to provide spectra in a wider wavelength range and subtract the baseline.
8. There are some typos in the manuscript. For example, a space was missing between “0V” on Page 12. On Page 10, Fig. S8 was cited as a reference for Cu-EDA absorbance, while Fig. S8 showed the schematic illustration of spectro-electrochemical configuration.

Reviewer #3

(Remarks to the Author)

The manuscript addresses an important bottleneck in electrochemical carbon capture—namely, the energy cost associated with amine regeneration via EMAR. The integrated use of electrochemical techniques, spectroscopy, and simulations is commendable and provides compelling evidence for the role of anion identity in modulating Cu redox behavior and CO₂ release efficiency.

To further strengthen the manuscript, I would encourage authors to consider the following questions and suggestions for clarification before the manuscript is accepted in Nature Communications.

1. Have the authors explored how varying the concentration of Cl⁻, NO₃⁻, or ClO₄⁻ affects the observed trends in CO₂ release potential and efficiency? This could determine whether the effect is general or specific to high ionic strength.
2. Can the authors provide more direct evidence of Cu oxidation state or surface speciation (e.g., XPS, in situ Raman, EQCM) during the EMAR process across different anions? Especially since Cl⁻ may alter the Cu surface in ways that impact both redox behavior and amine complexation.
3. I wonder whether any Cl⁻-induced side reactions (e.g., Cl₂ evolution or CuCl precipitation) were observed or considered? How stable is the Cu electrode under prolonged cycling in chloride-containing electrolyte?
4. Do authors expect these findings to extend to other amine systems (e.g., sterically hindered or monoamines)? Please check Investigating proton shuttling and electrochemical mechanisms of amines in integrated CO₂ capture and utilization. Nat Commun 15, 9207 (2024). DOI: 10.1038/s41467-024-53543-4

In the present manuscript, their findings are centered on ethylenediamine (EDA), which is a small, unhindered diamine forming stable complexes with Cu²⁺. Could you comment on whether the Cl⁻-induced enhancement in CO₂ release is expected to generalize to:

- Sterically hindered diamines or longer chain diamines with reduced complexation ability?
- Monoamines (e.g., MEA, DEA) commonly used in practical CO₂ capture?
- Tertiary amines, where bicarbonate rather than carbamate formation dominates?

5. Can authors clarify whether Cl⁻ primarily enhances Cu²⁺ formation or whether it also modifies the amine–carbamate equilibrium or Cu–amine complex stability? It would be interesting to disentangle these effects.

6. Have you considered whether this anion-modulated EMAR process could be integrated with downstream electrochemical CO₂ conversion (e.g., to formate or CO)? Some insight into compatibility or limitations would broaden the impact.

7. Last, the authors state that this is the first time-resolved, in situ characterization of interfacial molecular processes during EMAR-based CO₂ release. While the level of integration across techniques (DEMS, UV-vis, ATR-FTIR, MD) is indeed novel and well-executed, I encourage the authors to clarify the scope of their claim and differentiate it from earlier EMAR studies that employed in situ or operando characterization. Explicitly stating what aspect is being reported “for the first time” (e.g., time-resolved correlation of Cu oxidation with CO₂ desorption dynamics across electrolyte systems) would sharpen the manuscript’s contribution and avoid unintended overstatement. Examples of work:

- Liu et al., Nature Energy, 2021
- Title: “Electrochemically mediated direct air capture of CO₂ using quinone chemistry in salt-concentrated media”
- Subraveti et al., ACS Energy Letters, 2022
- Title: “Electrochemical CO₂ Capture from Amine Solutions: The Impact of Solvent and Current Density”
- Zhang et al., ChemSusChem, 2021
- Title: “Electrochemical Regeneration of Spent Amine Solvent for Energy-Efficient CO₂ Capture”
-

Version 1:

Reviewer comments:

Reviewer #1

(Remarks to the Author)

The authors have modified the manuscript carefully on the basis of the comments of the reviewers. I think the revised manuscript can be accepted.

Reviewer #2

(Remarks to the Author)

The authors have adequately addressed most of the concerns, and the revised manuscript shows improved quality. However, a few issues still require further discussion.

RE 3: The authors agree that the FE loss should be attributed to the formation of free Cu²⁺ and Cu⁺ compounds, rather than to the formation of [Cu(EDA)₂]²⁺. However, on page 8 of the manuscript, it is still stated: “The limited faradaic losses in Fig.

S3 may be related to free amine molecules in the feed, which will lead to undesired direct Cu oxidation and $[\text{Cu}(\text{EDA})_2]^{2+}$ formation." This statement appears inconsistent with the authors' response and should be clarified.

RE 5: The authors selected the testing potential range based on the open-circuit potential (OCP). Should the overpotential comparison therefore be made relative to OCP rather than to the Ag/AgCl reference electrode?

RE 7: In Fig. N4, the baseline for the 0.1 M KCl + EDA-CO₂ data appears to be sloping, as the absorbance at 400 nm is close to 0, while it reaches approximately 0.05 at 800 nm. Could the authors provide absorbance curves over a broader wavelength range to better determine the baseline?

Reviewer #3

(Remarks to the Author)

The authors have addressed the reviewers' comments thoroughly and have significantly strengthened their manuscript with additional experiments, analyses, and clarifications. The new data—including expanded control studies, broader amine scope, baseline corrections, and mechanistic clarifications—further reinforce the robustness and generality of the findings.

The work convincingly demonstrates the critical role of electrolyte anions, particularly chloride, in lowering the energy requirements of electrochemically mediated amine regeneration for CO₂ capture. The integration of operando spectroscopy, mass spectrometry, and molecular dynamics simulations provides a uniquely comprehensive mechanistic picture. The study not only advances our fundamental understanding of EMAR processes but also offers practical insights into electrolyte design for more energy-efficient CO₂ capture technologies.

Given the originality, methodological rigor, and potential impact across electrochemistry, carbon capture, and energy research, I recommend this revised manuscript for publication in Nature Communications.

Version 2:

Reviewer comments:

Reviewer #2

(Remarks to the Author)

Authors have addressed all my comments and acceptance is now recommended.

Responses to the Reviewers' comments

The entire comments from Reviewer #1

Reviewer #1 (Remarks to the Author): This manuscript presents novel and significant findings on the influence of electrolyte anions, particularly Cl^- , on electrochemically mediated amine regeneration (EMAR) CO_2 capture, supported by a well-integrated experimental characterizations and molecular dynamics simulations. However, some key mechanistic aspects require further clarification or expansion to fully establish the scientific rigor and generality of the conclusions. I think the manuscript can become acceptable after major revision.

Overall response: We greatly appreciate Reviewer 1's constructive suggestions as well as concerns guiding the revision of our manuscript.

Point-to-point Response to Reviewer 1:

Comment 1: *How exactly does Cl^- interact with the Cu surface? Direct evidence should be included to confirm the formation of Cu–Cl complex. What is the interaction between Cu and NO_3^- or ClO_4^- , can Cu– NO_3 or Cu– ClO_4 complexes form?*

Response 1: We thank Reviewer 1 for the thoughtful comments. Voltammetry, as a surface-sensitive technique, reveals distinct responses from adsorbed species, making it a powerful tool to provide direct evidence for Cu–Cl complex formation (Bard, Allen J., and Larry R. Faulkner. *Electrochemical Methods: Fundamentals and Applications*. 2nd ed., Wiley, 2001.); the peak at 0.08 V followed by a sharp current drop supports a dissolution and precipitation mechanism, suggesting that the diffusive species is CuCl_2^- , formed via $\text{Cu}^{2+} + 2\text{Cl}^- \rightarrow \text{CuCl}_2^- + \text{e}^-$. (B. V. Tilak, R. S. Perkins, H. A. Kozłowska, B. E. Conway, *Electrochimica Acta* 17, 1447, 1972; J. Crousier, L. Pardessus, J. -P. Cousier, *Electrochimica Acta*, 33, 8, 1039, 1988).

Moreover, Cu can interact with nitrate (NO_3^-) to form coordination complexes through Cu–O bonds, with nitrate binding in monodentate, bidentate, or bridging modes depending on the coordination environment. In contrast, perchlorate (ClO_4^-) is a weakly coordinating anion that typically acts as a non-coordinating counterion due to its large size and low basicity. While Cu– NO_3 complexes are chemically meaningful, the bonding strength is only moderate. As a result, in EMAR systems, NO_3^- has limited influence on performance, it may slightly affect the redox behavior or interfacial structure but does not strongly compete with amines or disrupt the CO_2 capture cycle.

Action: To give the readers more clear information, we have added the corresponding explanations “Notably, cyclic voltammetry in Fig. 2A, conducted in 0.1 M KCl with EDA– CO_2 electrolyte, showed a distinct oxidation peak at ~ 0.08 V vs. Ag/AgCl followed by a consistent sharp current drop, indicating that the electrode surface formed via a dissolution–precipitation mechanism, with the diffusing species likely being the CuCl_2^- complex.” and the corresponding literature in the revised manuscript.

Comment 2: *The authors focus exclusively on the anodic CO_2 release step, but the initial CO_2 capture*

via carbamate formation (e.g., $\text{CO}_2 + \text{EDA} \rightarrow \text{EDA-CO}_2^-$) is equally critical to the EMAR process. Do different anions (e.g., Cl^-) influence the equilibrium or kinetics of CO_2 binding to EDA (or dimethylamine, if used)? If anion identity affects the speciation or solubility of amine- CO_2 adducts, this could alter the initial concentrations for electrochemical steps, complicating mechanistic interpretation. A discussion, or ideally, experimental data, on anion effects during carbamate formation would significantly strengthen the evaluation of the full electrochemical cycle.

Response 2: That is a valid point. We agree that the nature of the supporting anion could potentially influence the equilibrium and kinetics of amine- CO_2 adduct formation. To examine this possibility, we conducted control experiments in which CO_2 was first introduced into an EDA solution, followed by the addition of different supporting salts. Notably, no CO_2 release was observed upon salt addition, suggesting that the presence of Cl^- or other charged entities in the solution did not significantly disrupt the carbamate equilibrium under our experimental conditions. This indicates that the speciation of amine- CO_2 adducts remains stable and that variations in anion identity are unlikely to affect the initial concentrations of captured CO_2 relevant to the electrochemical step.

Comment 3: All experiments and simulations in this study employ EDA, a bidentate ligand known for its strong Cu^{2+} chelation. However, would the observed Cl^- -promoted Cu^{2+} release and CO_2 desorption hold for other amines? If the authors have tested alternative amines experimentally or via MD simulations, such data would greatly strengthen the mechanistic understanding. If not, this should be explicitly acknowledged as a limitation, along with a brief discussion on whether the Cl^- effect is likely system-specific (e.g., dependent on bidentate coordination).

Response 3: We thank Reviewer 1 for this valuable comment. We fully agree that testing a broader range of amines would greatly enhance the mechanistic understanding of the EMAR process. In this study, we selected EDA because it is one of the most widely studied and commonly used amines in EMAR systems, owing to its high CO_2 absorption capacity and efficient electrochemical regeneration.

From a fundamental perspective, the Cl^- -promoted CO_2 release is primarily driven by the direct interaction between Cl^- and Cu^{2+} , which facilitates the displacement of Cu^{2+} from the amine-Cu complex and subsequently triggers CO_2 desorption. While the amine structure may influence the local coordination environment and modulate the extent of this effect, the dominant factor remains the anion-metal interaction.

In line with the reviewer's suggestion and to probe the generality of this effect, we performed preliminary experiments using N-methyldiethanolamine (MDEA), a tertiary amine with weaker Cu^{2+} binding affinity and a distinct coordination mode. Interestingly, we observed similar Cl^- -promoted Cu^{2+} release and CO_2 desorption behavior, indicating that the anion effect is not limited to strongly chelating bidentate ligands like EDA. Although further systematic studies are warranted, these initial results suggest that the Cl^- effect may be broadly applicable across different classes of amines.

Fig. N1 (also Fig. S5 of the revised manuscript) CO₂ release-cyclic voltammetry for Cu foil in 0.1 M KA (A = Cl⁻, NO₃⁻ and ClO₄⁻) supporting electrolyte with 0.1 M N-methyldiethanolamine (MDEA)-CO₂ carbamates in H cell. All the CVs were recorded at 5 mV s⁻¹.

Action: We have added the CO₂ release-cyclic voltammetry on Cu in MDEA (Fig. N1) and the corresponding statements “As shown in Fig. S5, similar Cl⁻-promoted Cu²⁺ release and CO₂ desorption were also observed using N-methyldiethanolamine (MDEA), a tertiary amine with weaker Cu²⁺ binding affinity. This suggests that the anion effect could potentially extend to other classes of amines with varying coordination strengths and structures.” in the revised supporting information.

Comment 4: *The UV-vis spectroscopy at 550 nm was used to monitor the formation of [Cu(EDA)₂]²⁺, and this was taken as an indirect measure of CO₂ release. Any direct evidence for confirming this complex formation?*

Response 4: We thank Reviewer 1 for raising this important point. The UV-vis absorbance at 550 nm is characteristic of the d-d transition of the [Cu(EDA)₂]²⁺ complex and has been widely used in previous studies as a spectroscopic signature for its formation. To support this assignment, we compared the UV-vis spectra of standard solutions of [Cu(EDA)₂]²⁺ prepared in the absence of CO₂ and observed a consistent absorption peak at 550 nm, matching the spectra obtained during electrochemical CO₂ release.

Fig. N2 (also Fig. S9A of the revised manuscript) UV-Vis spectra for Cu²⁺, [Cu(EDA)]²⁺, and [Cu(EDA)₂]²⁺, respectively in Cu-EDA system.

Action: To provide clearer evidence for the complex formation, we have added the UV–vis spectrum of the standard $[\text{Cu}(\text{EDA})_2]^{2+}$ solution shown as Fig. N2 and the corresponding statements “UV-vis spectra showed absorption bands at 820, 650, and 550 nm in Cu-EDA and Cu-EDA- CO_2 system, respectively, which could be attributed to Cu^{2+} , $[\text{Cu}(\text{EDA})]^{2+}$, and $[\text{Cu}(\text{EDA})_2]^{2+}$, respectively.” in the revised supporting information.

Comment 5: *The results show that that CO_2 releases faster in the presence of KCl. However, MD simulations demonstrated that anion regulation promotes the preferential accumulation of neutral CO_2 and carbamate species near the Cu interface in the Cu-Cl system, facilitating Cu^{2+} ion formation and resulting in lower CO_2 oxidation onset potentials. The authors attempt to explain kinetic observations using thermodynamic arguments, which represents a fundamental conceptual mismatch. A rigorous kinetic analysis would be required to properly interpret these reaction dynamics.*

Response 5: We would like to thank Reviewer 1 for this thoughtful and important comment. We fully agree that our molecular dynamics (MD) simulations primarily reveal the distinct local structural distributions of neutral CO_2 and carbamate species near the Cu interface in different anionic environments under stationary conditions. The purpose of these simulations is to provide mechanistic insight into the pre-reactive interfacial states that could influence electrochemical behavior. Specifically, our results show that in the presence of Cl^- , the interfacial accumulation of CO_2 and carbamate species containing species is significantly enhanced, and more Cl^- anions aggregate at the Cu surface. The locally anion enriched environment is expected to shift the thermodynamics of the $\text{Cu}^+/\text{Cu}^{2+}$ formation (c.f. Refs. 28, 29, 32, 33, 42). To show this we consider the Cl^- assisted Cu oxidation according to “ $\text{Cu}^0 + 2 \text{Cl}^- \rightarrow \text{CuCl}_2 + 2 \text{e}^-$ ” and formulate its thermodynamic Nernst expression which reads $E(\text{Cu}/\text{CuCl}_2) = E^0(\text{Cu}/\text{CuCl}_2) + RT/2F \ln ([\text{CuCl}_2]/[\text{Cl}^-]^2)$. As Cl^- becomes locally enriched at the interface, the thermodynamic onset potential of this redox process shifts cathodically, such that the Cu oxidation (and hence the CO_2 release) appears earlier at a given electrode potential on the SHE scale. This is consistent with the experimentally observed earlier onset of CO_2 release. So, here thermodynamics indirectly affects the apparent reaction rates. We have clarified this interpretation of our computational results in the manuscript.

Action: To this end, we have revised the original sentence “Local enrichment of Cl^- at the Cu surface shifts the thermodynamic $\text{Cu}^0/\text{Cu}^{2+}$ redox potential cathodically, leading to Cu oxidation at lower potentials and thus an earlier onset of CO_2 release.” “Strong Cu-Cl interaction with chloride anion accumulations could facilitate Cu oxidation at more negative potentials, which could promote the subsequent CO_2 release dynamics (Fig. S15).” to avoid any conceptual ambiguity.

Comment 6: *The manuscript contains several formatting inconsistencies. For example, “ Cl^- ” (with a proper superscript minus sign) is the correct notation, whereas “Cl-” (with a hyphen) is incorrect. The author has used both forms interchangeably. Please double-check the manuscript.*

Response 6: We would like to thank Reviewer 1 for correctly pointing this out. We have modified the text accordingly.

Action: We have replaced the expressions of “Cl” to “Cl⁻” in the revised manuscript and supporting information.

The entire comments from Reviewer #2

Reviewer #2 (Remarks to the Author): The authors proposed a Cl-mediated EMAR CO₂ release pathway, which reported a lower CO₂ release onset potential. Time-resolved DEMS, in situ UV-vis, in situ FTIR, and MD calculations were combined to provide relatively complete evidence to clarify the importance of local anion microenvironment in EMAR systems. The research focus of this work is interesting and meaningful in EMAR research, while the data analysis should be strengthened to make the conclusion more persuasive.

Overall response: We greatly appreciate Reviewer 2 for the feedback.

Point-to-point Response to Reviewer 2:

Comment 1: *Could the authors provide kinetics or thermodynamics data about the complex reaction between Cu⁺²⁺ and Cl⁻, and compare it with that of Cu⁺²⁺ and EDA? As shown in Equation (3), the formation of Cu-Am complex is already a very thermodynamically preferred reaction. If the coordination stability of Cu⁺²⁺ and Cl⁻ is stronger, would the formation of Cu-Cl complex hinder the release of CO₂?*

Response 1: We thank Reviewer 2 for this insightful question. As noted in Equation (3), the formation of the Cu-amine (Cu-Am) complex is thermodynamically favorable, with a high stability constant ($\log K_{\text{Cu(Am)}_n} \approx 18-20$), reflecting the strong chelating ability of EDA. In contrast, the interaction between Cu²⁺ and Cl⁻ is significantly weaker, with reported log K values in the range of 0.7–1.5 for the initial coordination step. (D. F. C. Morris, E. L. Short, J. Chem. Soc., 1962, 2672; Y. Meng, A. J. Bard, Anal. Chem. 2015, 87, 3498)

In our system, the role of Cl⁻ is not to form a more stable complex with Cu²⁺ than the amine, but rather to facilitate the release of Cu²⁺ from the Cu-amine complex at the early stage of the anodic step. Our results suggest that Cl⁻ can transiently occupy part of the Cu²⁺ coordination sphere, weakening the Cu-EDA interaction and thereby promoting CO₂ desorption. Since the Cu-Cl coordination is not thermodynamically dominant, it does not hinder CO₂ release; instead, it promotes dynamic ligand exchange that enhances the efficiency of electrochemical regeneration.

Comment 2: *There was an oxidation peak at 0.08 V_{Ag/AgCl} only in KCl + EDA-CO₂ electrolyte as shown in Fig. 2A. What is the physics chemical process corresponding?*

Response 2: We thank Reviewer 2 for the thoughtful comments. As mentioned in Response 1 to Reviewer 1, the peak at 0.08 V followed by a sharp current drop indicates that the film was formed by a mechanism of dissolution-precipitation, suggesting that the diffusive species is CuCl₂⁻, formed via $\text{Cu}^{2+} + 2\text{Cl}^- \rightarrow \text{CuCl}_2^- + \text{e}^-$. (B. V. Tilak, R. S. Perkins, H. A. Kozłowska, B. E. Conway, Electrochimica. Acta 17, 1447, 1972; J. Crousier, L. Pardessus, J. -P. Cousier, Electrochimica Acta, 33, 8, 1039, 1988).

Comment 3: *The authors attributed the loss of FE in Fig. S3 to direct Cu oxidation and [Cu(EDA)2]2+,*

which is confusing. According to Methods, FE means the ratio of charge used to form $[\text{Cu}(\text{EDA})_2]^{2+}$ and total charge, which seems to imply the lost charge was used to form Cu^{2+} , Cu-Cl complex, or Cl_2 , as no O_2 evolution was detected.

Response 3: Thanks for Reviewer 2's concerns. Approximately 10% of the charge appears to be consumed in competing processes near the double layer, which likely include the formation of free Cu^{2+} (which will be very small due to the large equilibrium constant), solid and dissolved Cu^+ compounds, such as Cu_2O or Cu-Cl, as the reviewer correctly pointed out. As shown in the DEMS data in Fig. S5, no O_2 signal was detected. Moreover, the applied anodic potentials remain below the standard redox potential for chloride oxidation ($E^\circ = +1.36 \text{ V}$ vs. RHE at pH 0, and even more positive E° values at alkaline pH values). Therefore, the generation of both O_2 and Cl_2 can be excluded under the investigated conditions.

Comment 4: The authors declared their Cl-mediated interfacial EMAR CO_2 release pathway enables "one of the lowest ever reported onset potentials of CO_2 release", which should be supported by data from other works.

Response 4: We would like to thank Reviewer 2 for the kind suggestions.

Action: We have incorporated the following references into the revised manuscript: Rahimi M, Diederichsen K. M., Ozbek N., Wang M., Choi W., Hatton T. A. *Environ Sci Technol.* 2020, 54, 8999; Hassan, A., Afshari, M., Rahimi, M. A. *Nat Commun* **16**, 2025, 6333; Wang M., Hariharan S., Shaw R. A., Hatton T. A. *International Journal of Greenhouse Gas Control*, 2019, 82, 48.

Comment 5: Please give a clear statement of potential ranges in captions of Figs. 3A-C. What is the reason to choose different potential ranges for three electrolytes? Would a more negative potential range used in KCl + EDA- CO_2 cause a more negative onset potential?

Response 5: We thank Reviewer 2 for kind suggestions. The potential windows for each electrolyte were selected relative to their respective open-circuit potentials (OCP), specifically within a range of approximately 0 to +0.5 V vs. OCP. Plotting the voltammogram vs the OCP does not do justice to earlier onset potentials, however, this approach allows for a more direct inspection and comparison of the current-potential characteristics of the Cu oxidation across different electrolytes by aligning their thermodynamic equilibrium potentials. In order to show onset potentials benefits, we prefer plotting the voltammetry on the SHE or RHE scales.

Action: We have now added the corresponding statement of the applied potential ranges in the captions of Figs. 3A-C in revised manuscript, as below, "The potential windows for each electrolyte were selected relative to their respective open-circuit potentials (OCP)".

Comment 6: The authors attributed the decreasing FTIR intensity of free CO_2 during the cathodic scan to the decreasing reaction rate. Why is the signal of CO_2 not continuously accumulating?

Response 6: We thank Reviewer 2 for this thoughtful question. The observed decrease in the FTIR signal of free CO_2 during the cathodic scan is attributed to mass transport processes. During the anodic potential scan, the applied potential controls the anodic Cu oxidation rate

and indirectly the CO₂ release. As a result of this, the local CO₂ concentration increases with potential. However, at later stages during the anodic scan, diffusional and (free) convective CO₂ mass transport away from the interfacial region into the bulk of the FTIR cell becomes noticeable and prevents a continuous accumulation of local interfacial CO₂ concentration. As the CO₂ release rate drops exponentially during the cathodic Cu deposition the interplay of reduced CO₂ release rate and mass transport results in a net decrease of local CO₂. This is reflected in the diminishing FTIR signal.

Comment 7: The discussion about the effects of baseline or data in the background was missing in several experiments:

a. The Am- CO₂ electrolyte was obtained after CO₂ flow overnight at a flow rate of 10 sccm, which is likely to form a CO₂-over-saturated solution. Would it lead to a continuous CO₂ release without any electrochemical experiments? Should this spontaneous CO₂ release be excluded as a baseline when analyzing data?

b. In Figs. 3A-3C, three current signals showed different trends: the current signal in 3A is very stable, it in 3B starts almost from 0, while it in 3C shows a decreasing trend. Would it be related to the baseline in the measurements?

c. In Fig. S10, it is suggested to provide spectra in a wider wavelength range and subtract the baseline.

Response 7: We thank Reviewer 2 for pointing this out and have removed the baselines from Figs. 3A–C and S10 accordingly.

Fig. N3 (also Fig. 3 of the revised manuscript) Differential Electrochemical Mass Spectrometry (DEMS) studies of the anion impact on EMAR interfacial CO₂ release kinetics.

Fig. N4 (also Fig. S12 of the revised manuscript) In situ UV-Vis on Cu NPs in different electrolyte after CO₂ release.

Action: We have replaced the DEMS CO₂ signal and the in situ UV-vis spectra in Fig. 3 and Fig. S10, respectively, with baseline-corrected data.

Comment 8: *There are some typos in the manuscript. For example, a space was missing between “0V” on Page 12. On Page 10, Fig. S8 was cited as a reference for Cu-EDA absorbance, while Fig. S8 showed the schematic illustration of spectro-electrochemical configuration.*

Response 8: We would like to thank Reviewer 2 for this comment. We have modified the text accordingly.

The entire comments from Reviewer #3

Reviewer #3 (Remarks to the Author): The manuscript addresses an important bottleneck in electrochemical carbon capture—namely, the energy cost associated with amine regeneration via EMAR. The integrated use of electrochemical techniques, spectroscopy, and simulations is commendable and provides compelling evidence for the role of anion identity in modulating Cu redox behavior and CO₂ release efficiency.

To further strengthen the manuscript, I would encourage authors to consider the following questions and suggestions for clarification before the manuscript is accepted in Nature Communications.

Overall response: We appreciate Reviewer 3’s helpful input and thoughtful concerns, which have helped guide the revision of our manuscript.

Point-to-point Response to Reviewer 3:

Comment 1: *Have the authors explored how varying the concentration of Cl⁻, NO₃⁻, or ClO₄⁻ affects the observed trends in CO₂ release potential and efficiency? This could determine whether the effect is general or specific to high ionic strength.*

Response 1: We thank Reviewer 3 for this valuable suggestion. According to Reviewer 3’s suggestion, we conducted experiments to evaluate the effect of varying Cl⁻, NO₃⁻, and ClO₄⁻ concentrations at 0.07 M and 0.1 M. The results indicate that the anion concentration has only a minimal impact on CO₂ release behavior.

Fig. N5 (also Fig. S4 of the revised manuscript) Cyclic voltammetry of Cu foil in 0.1 M and 0.07 M KA (A = Cl⁻, NO₃⁻, and ClO₄⁻) supporting electrolytes containing 0.1 M EDA-CO₂ carbamates. All the CVs were recorded at 5 mV s⁻¹.

Action:

We have added the cyclic voltammograms of Cu foil in 0.1 M and 0.07 M KA (A = Cl⁻, NO₃⁻, and ClO₄⁻) supporting electrolytes containing 0.1 M EDA-CO₂ carbamates (Fig. N5), along with the statement: “0.1 M and 0.07 M supporting electrolytes were used for the measurements, and the anion concentration showed only a minimal impact on the CO₂ release behavior.” in the revised supporting information.”

Comment 2: *Can the authors provide more direct evidence of Cu oxidation state or surface speciation (e.g., XPS, in situ Raman, EQCM) during the EMAR process across different anions? Especially since Cl⁻ may alter the Cu surface in ways that impact both redox behavior and amine complexation.*

Response 2: We would like to thank Reviewer 3 for the insightful suggestion. We agree that direct evidence of Cu oxidation states and surface speciation would further strengthen the mechanistic understanding of our system. we acknowledge the importance of surface-sensitive techniques such as XPS, in situ Raman, or EQCM. These methods are part of our ongoing investigations and will be pursued in future work to complement the current findings and elucidate possible anion-dependent surface modifications. Moreover, as discussed in our Response 1 to Reviewer 1, electrochemical techniques, especially cyclic voltammetry, are highly sensitive to surface coordination changes and interfacial processes. The distinct electrochemical responses observed in the presence of Cl⁻, including altered oxidation behavior and decreased Faradaic efficiency, are consistent with the formation of Cu-Cl species, as supported by prior literature. While direct spectroscopic confirmation remains desirable, the electrochemical evidence strongly suggests that Cu-Cl coordination occurs under our experimental conditions.

Comment 3: *I wonder whether any Cl⁻-induced side reactions (e.g., Cl₂ evolution or CuCl precipitation) were observed or considered? How stable is the Cu electrode under prolonged cycling in chloride-containing electrolyte?*

Response 3: We would like to thank Reviewer 3 for the comment. Cl₂ evolution is unlikely

under our experimental conditions, as the applied anodic potentials remain well below the standard redox potential for Cl^- oxidation (+1.36 V vs. RHE at pH 0). While Cl^- can facilitate a mechanism of dissolution-precipitation involving the formation of diffusive species CuCl_2^- , via $\text{Cu}^{2+} + 2\text{Cl}^- \rightarrow \text{CuCl}_2^- + \text{e}^-$. In fact, since the Cu electrode undergoes continuous oxidation to Cu^{2+} during EMAR cycling, thus Cu electrode itself is not inherently stable under these conditions.

Comment 4: *Do authors expect these findings to extend to other amine systems (e.g., sterically hindered or monoamines)? Please check Investigating proton shuttling and electrochemical mechanisms of amines in integrated CO_2 capture and utilization. Nat Commun 15, 9207 (2024). DOI: 10.1038/s41467-024-53543-4*

In the present manuscript, their findings are centered on ethylenediamine (EDA), which is a small, unhindered diamine forming stable complexes with Cu^{2+} . Could you comment on whether the Cl^- -induced enhancement in CO_2 release is expected to generalize to:

- *Sterically hindered diamines or longer chain diamines with reduced complexation ability?*
- *Monoamines (e.g., MEA, DEA) commonly used in practical CO_2 capture?*
- *Tertiary amines, where bicarbonate rather than carbamate formation dominates?*

Response 4: We thank Reviewer 3 for this comment. We fully agree that extending our study to other amine systems would enhance the significance and generality of the findings.

In this work, we selected EDA because it is one of the most widely studied and commonly used amines in EMAR systems, owing to its high CO_2 absorption capacity and efficient electrochemical regeneration. From a mechanistic perspective, the Cl^- -promoted CO_2 release is primarily driven by the direct interaction between Cl^- and Cu^{2+} , which facilitates the displacement of Cu^{2+} from the amine–Cu complex and subsequently triggers CO_2 desorption. While the amine structure may influence the local coordination environment and modulate the extent of this effect, the dominant factor remains the anion–metal interaction.

As mentioned in the Response 3 to Reviewer 1, and according to Reviewer 3’s kind suggestion, we performed preliminary experiments using N-methyldiethanolamine (MDEA), a tertiary amine with weaker Cu^{2+} binding affinity and a distinct coordination mode. Interestingly, we observed similar Cl^- -promoted Cu^{2+} release and CO_2 desorption behavior, indicating that the anion effect is not limited to strongly chelating bidentate ligands like EDA. Although further systematic studies are warranted, these initial results suggest that the Cl^- effect may be broadly applicable across different classes of amines.

Action: We have cited the suggested literature (*Nat. Commun.* **15**, 9207 (2024)) in the revised manuscript. We have also added the CO_2 release-cyclic voltammetry on Cu in MDEA (Fig. N1) and the corresponding statements “As shown in Fig. S5, similar Cl^- -promoted Cu^{2+} release and CO_2 desorption were also observed using N-methyldiethanolamine (MDEA), a tertiary amine with weaker Cu^{2+} binding affinity. This suggests that the anion effect could potentially extend to other classes of amines with varying coordination strengths and structures.” in the revised supporting information.

Comment 5: *Can authors clarify whether Cl^- primarily enhances Cu^{2+} formation or whether it also modifies the amine–carbamate equilibrium or Cu–amine complex stability? It would be interesting to disentangle these effects.*

Response 5: We thank Reviewer 3 for this thoughtful comment. Based on our molecular dynamics simulations and supporting experimental observations, the primary role of Cl^- appears to be enhancing Cu^{2+} formation through strong interfacial Cu–Cl interactions. This facilitates the oxidation of Cu and promotes the release of CO_2 from the carbamate complex.

While Cl^- significantly alters the interfacial electric field and promotes local accumulation of carbamate and CO_2 molecules near the Cu surface, we did not observe evidence suggesting that Cl^- directly shifts the amine–carbamate equilibrium or destabilizes the Cu–amine complex in bulk solution. Rather, the Cl^- effect is interfacial and electrochemical in nature, accelerating Cu^{2+} generation at the electrode surface, which in turn drives CO_2 release. Disentangling these coupled processes is indeed complex, and we agree that more detailed thermodynamic and kinetic studies—particularly probing speciation and stability constants—would be valuable in future work.

Comment 6: *Have you considered whether this anion-modulated EMAR process could be integrated with downstream electrochemical CO_2 conversion (e.g., to formate or CO)? Some insight into compatibility or limitations would broaden the impact.*

Response 6: Thank you for the insightful suggestion. The integration of the anion-modulated EMAR process with downstream electrochemical CO_2 conversion is an important direction. We did not consider it in this current contribution, but we are aware of its future significance and of its challenges related to the release being an anodic process, while the conversion being a cathodic one. Here we tried to address interfacial characterization and a more fundamental kinetic understanding of the CO_2 release step.

Action: As suggested by Reviewer 3, we have highlighted the potential of integrating the EMAR process with downstream CO_2 electroreduction. The revised manuscript now includes the following statement: “Our findings on the anion effect offer a facile strategy to lower regeneration energy consumption in NETs and DAC, while our insights into electrolyte compatibility and interfacial engineering enable localized, controllable CO_2 release, providing a foundation for integration with downstream CO_2 conversion.”

Comment 7: *Last, the authors state that this is the first time-resolved, in situ characterization of interfacial molecular processes during EMAR-based CO_2 release. While the level of integration across techniques (DEMS, UV-vis, ATR-FTIR, MD) is indeed novel and well-executed, I encourage the authors to clarify the scope of their claim and differentiate it from earlier EMAR studies that employed in situ or operando characterization. Explicitly stating what aspect is being reported “for the first time” (e.g., time-resolved correlation of Cu oxidation with CO_2 desorption dynamics across electrolyte systems) would sharpen the manuscript’s contribution and avoid unintended overstatement. Examples*

of work:

- Liu et al., *Nature Energy*, 2021

- Title: “Electrochemically mediated direct air capture of CO₂ using quinone chemistry in salt-concentrated media”

- Subraveti et al., *ACS Energy Letters*, 2022

- Title: “Electrochemical CO₂ Capture from Amine Solutions: The Impact of Solvent and Current Density”

- Zhang et al., *ChemSusChem*, 2021

- Title: “Electrochemical Regeneration of Spent Amine Solvent for Energy-Efficient CO₂ Capture”

Response 7: We thank Reviewer 3 for this thoughtful comment. The first referenced work (Liu et al., *Nature Energy*, 2021) focuses on quinone-mediated CO₂ capture, which operates via a fundamentally different mechanism from EMAR and does not involve metal–amine complexation. The latter two references (Subraveti et al., *ACS Energy Lett.*, 2022; and Zhang et al., *ChemSusChem*, 2021) seem not to exist in the cited journals. We would be grateful if the reviewer could kindly provide and we would be glad to cite and discuss them accordingly.

To the best of our knowledge, this work represents the first report of time-resolved, in situ characterization of interfacial molecular processes specifically during EMAR-based CO₂ release.

Responses to the Reviewers' comments

The entire comments from Reviewer #1

Reviewer #1 (Remarks to the Author): The authors have modified the manuscript carefully on the basis of the comments of the reviewers. I think the revised manuscript can be accepted.

Overall response: We greatly appreciate the Reviewer 1 for the positive evaluation and kind recommendation for acceptance.

The entire comments from Reviewer #2

Reviewer #2 (Remarks to the Author): The authors have adequately addressed most of the concerns, and the revised manuscript shows improved quality. However, a few issues still require further discussion.

Overall response: We appreciate Reviewer 2's helpful input and thoughtful concerns, which have helped guide the revision of our manuscript. We address the remaining concerns in detail below.

Point-to-point Response to Reviewer 2:

Comment 1: RE 3: The authors agree that the FE loss should be attributed to the formation of free Cu^{2+} and Cu^+ compounds, rather than to the formation of $[\text{Cu}(\text{EDA})_2]^{2+}$. However, on page 8 of the manuscript, it is still stated: "The limited faradaic losses in Fig. S3 may be related to free amine molecules in the feed, which will lead to undesired direct Cu oxidation and $[\text{Cu}(\text{EDA})_2]^{2+}$ formation." This statement appears inconsistent with the authors' response and should be clarified.

Response 1: We thank the reviewer for noting this point. The statement on page 8 has been revised to clarify the FE loss.

Action: We have modified the statements as below: "The limited faradaic losses in Fig. S3 may be related to competing processes near the double layer, which likely include the formation of Cu^{2+} and Cu-Cl species. While presence of free EDA in the feed would also result in inefficiencies due to $[\text{Cu}(\text{EDA})_2]^{2+}$ complex formation, this inefficiency route is likely not dominant under our CO_2 saturated carbamate feed conditions."

Comment 2: RE 5: The authors selected the testing potential range based on the open-circuit potential (OCP). Should the overpotential comparison therefore be made relative to OCP rather than to the Ag/AgCl reference electrode?

Response 2: We thank the reviewer for this valuable comment. According to the reviewer's suggestion, we have added Table S1 comparing the onset potentials versus Ag/AgCl and OCP in various supporting electrolytes. We have also added an explanation in the FTIR and UV-Vis sections to clarify the basis for the selected potential range.

Action: We have added Table N1 in the revised manuscript. In addition, we have modified the statements as follows: "UV-Vis spectra were collected in 0.1 M KCl + EDA- CO_2 electrolyte approximately every 2 s under potential sweeps from -0.24 to $+0.36 \text{ V}_{\text{Ag/AgCl}}$, corresponding to $0-0.6 \text{ V}_{\text{OCP}}$." and "FTIR spectra recorded during anodic voltammetric scans from OCP (-0.24

V to +0.26 $V_{\text{Ag/AgCl}}$, corresponding to 0–0.5 V_{OCP}) revealed a distinct vibrational feature at 2340 cm^{-1} (Fig. 4C, D, and S13), attributable to linear free CO_2 , confirming its evolution according to equation (3).”

Electrolyte	Onset potential	
	$V_{\text{Ag/AgCl}}$	V_{OCP}
KCl	-0.24	0.01
KNO_3	-0.12	0.06
KClO_4	-0.10	0.09

Table N1 (also Table S1 of the revised manuscript) Comparison of onset potentials versus Ag/AgCl and OCP in 0.1 M KA (A = Cl^- , NO_3^- , and ClO_4^-) supporting electrolytes.

Comment 3: RE 7: In Fig. N4, the baseline for the 0.1 M KCl + EDA- CO_2 data appears to be sloping, as the absorbance at 400 nm is close to 0, while it reaches approximately 0.05 at 800 nm. Could the authors provide absorbance curves over a broader wavelength range to better determine the baseline?

Response 3: We thank Reviewer 2 for pointing this out and have included absorbance spectra over a broader wavelength range in the revise Fig. S12 accordingly.

Action: We have replaced the in situ UV-vis spectra over a broader wavelength range in Fig. S12.

Fig. N1 (also Fig. S12 of the revised manuscript) In situ UV-Vis on Cu NPs in different electrolyte after CO_2 release.

The entire comments from Reviewer #3

Reviewer #3 (Remarks to the Author): The authors have addressed the reviewers’ comments thoroughly and have significantly strengthened their manuscript with additional experiments, analyses, and clarifications. The new data—including expanded control studies, broader amine scope, baseline corrections, and mechanistic clarifications—further reinforce the robustness and generality of the findings.

The work convincingly demonstrates the critical role of electrolyte anions, particularly chloride, in lowering the energy requirements of electrochemically mediated amine regeneration for CO₂ capture. The integration of operando spectroscopy, mass spectrometry, and molecular dynamics simulations provides a uniquely comprehensive mechanistic picture. The study not only advances our fundamental understanding of EMAR processes but also offers practical insights into electrolyte design for more energy-efficient CO₂ capture technologies.

Given the originality, methodological rigor, and potential impact across electrochemistry, carbon capture, and energy research, I recommend this revised manuscript for publication in Nature Communications.

Overall response: We sincerely thank Reviewer 3 for the thoughtful and encouraging feedback, and we greatly appreciate the recognition of our work's originality, rigor, and potential impact, as well as the recommendation for publication.